# Plasma Polymerization of Precipitated Silica for Tire Application

**DOI:** 10.3390/molecules28186646

**Published:** 2023-09-15

**Authors:** Sunkeun Kim, Wilma K. Dierkes, Anke Blume, Auke Talma, J. Ruud Van Ommen, Nicolas Courtois, Julian Davin, Carla Recker, Julia Schoeffel

**Affiliations:** 1Elastomer Technology and Engineering, University of Twente, 7500AE Enschede, The Netherlands; 2Sustainable Elastomer Systems, University of Twente, 7500AE Enschede, The Netherlands; 3Department of Chemical Engineering, Delft University of Technology, 2629HZ Delft, The Netherlands; j.r.vanommen@tudelft.nl; 4Continental Reifen Deutschland GmbH, 30419 Hannover, Germany

**Keywords:** plasma modification (PD), precipitated silica, natural rubber

## Abstract

Pre-treated silica with a plasma-deposited (PD) layer of polymerized precursors was tested concerning its compatibility with Natural Rubber (NR) and its influence on the processing of silica-silane compounds. The modification was performed in a tailor-made plasma reactor. The degree of deposition of the plasma-coated samples was analyzed by ThermoGravimetric Analysis (TGA). In addition, Diffuse Reflectance Infrared Fourier Transform spectroscopy (DRIFTs), X-ray Photoelectron Spectroscopy (XPS), and Transmission Electron Microscopy (TEM) were performed to identify the morphology of the deposited plasma polymer layer on the silica surface. PD silica samples were incorporated into a NR/silica model compound. NR compounds containing untreated silica and in-situ silane-modified silica were taken as references. The silane coupling agent used for the reference compounds was bis-(3-triethoxysilyl-propyl)disulfide (TESPD), and reference compounds with untreated silica having the full amount and 50% of silane were prepared. In addition, 50% of the silane was added to the PD silica-filled compounds in order to verify the hypothesis that additional silane coupling agents can react with silanol groups stemming from the breakdown of the silica clusters during mixing. The acetylene PD silica with 50% reduced silane-filled compounds presented comparable properties to the in-situ silane-modified reference compound containing 100% TESPD. This facilitates processing as lower amounts of volatile organic compounds, such as ethanol, are generated compared to the conventional silica-silane filler systems.

## 1. Introduction

Silica-silane is a commonly used filler system in tire technology, mainly for passenger car tire treads to reduce fuel consumption [1]. However, there are a few drawbacks to this technology: The processing of the silica-filled rubber compounds is elaborative [2], and toxic volatile organic compounds (VOCs) such as ethanol are generated during the mixing process [3]. Furthermore, this silica-silane system does not work efficiently for natural rubber, the main rubber used in truck tire treads. This is due to non-rubber components such as proteins and phospholipids [4]. One of the approaches to overcome these problems is to use pre-treated silica: The silica surface is modified to increase its hydrophobicity, improve dispersion, and finally form a bond with the polymer. Pre-treated silica eliminates the need for the addition of a coupling agent and reduces volatile organic compounds (VOCs) emissions during mixing [5,6,7,8].

Plasma polymerization deposition (PD) has been investigated as one of the technologies for surface modification of fillers without affecting the material’s bulk properties. It is possible to coat the surface of the filler using different monomers, which tailor the characteristics in terms of surface energy and functionality. Various studies have been conducted on the effect of plasma modification of a filler [9,10,11,12,13,14], including silica and carbon black, with different precursors to investigate the property enhancements in several rubbers and their blends. Nah et al. [9], improved the mechanical properties of SBR compounds by acetylene plasma-polymer deposition on silica. Tiwari et al. [10,11], studied the effect of various monomers, such as acetylene and thiophene, on the plasma treatment of precipitated silica for S-SBR and EPDM compounds. Plasma-modified silica with an appropriate monomer can improve the properties of compounds based on different polymers, such as dispersion and filler-polymer interaction. Mathew et al. [12], treated carbon black by plasma polymerization with acetylene as a precursor for controlled distribution in rubber blends.

This study investigates the effect of silica, pre-treated via plasma polymerization, on the properties of a NR/silica compound, for truck/bus tire tread applications. The literature-based screening of precursors for plasma treatment leads to acetylene as the first material of choice, which is expected to make the surface hydrophobic, thus enhancing the dispersibility in the NR matrix. In addition, unsaturated hydrocarbon structures as functional groups that remained in the deposited layer are expected to improve the rubber-filler interaction during vulcanization, forming chemical bonds by sulfur crosslinks. Firstly, the acetylene plasma-modified silica was characterized using ThermoGravimetric Analysis (TGA), Diffuse Reflectance Infrared Fourier Transform spectroscopy (DRIFTs), and X-ray Photoelectron Spectroscopy (XPS). Furthermore, NR compounds containing plasma-modified silica were studied. It is known that commercially available pre-treated silica has the disadvantage of unmodified silica surfaces being revealed during the mixing process when silica clusters are broken into smaller units [15]. Therefore, the effect of additional silane was investigated for the plasma-modified silica-filled NR compounds to cover the revealed new surface.

## 2. Results

### 2.1. Characterization of Plasma-Treated Silica

#### 2.1.1. Thermo-Gravimetric Analysis (TGA) Results

The weight loss of untreated and plasma-treated silica samples was measured to quantify the modification, and the normalized TGA curves are presented in Figure 1. The weight loss of the pure silica was 2.9% due to dehydration [16]. After acetylene plasma treatment, the weight loss increased to 7.0% for P1 and 9.3% for P2. In order to calculate the degree of deposition of the plasma-polymer-coated silica, the 2.9% humidity weight loss needs to be subtracted from the weight loss. Therefore, the degree of deposition of samples P1 and P2 was recalculated as 4.1% and 6.5%, respectively. When applied for the in-rubber test, these values will be compensated for the amount of plasma-treated silica.

Sample SP1 showed 5.1% weight loss, corresponding to 2.2% deposition of DAS, the S-containing precursor. SP2 showed 7.7% deposition after the first PD treatment with acetylene and an accumulated weight gain of 8.2% after the final treatment with DAS. The PD deposition of DAS resulted in an additional weight increase of 0.5%; such a comparatively low degree of deposition was chosen in order to prevent high sulfur levels in the deposited layer, which might cause pre-scorching during processing or over-curing.

Determining the actual structure of the deposited layer is a challenge due to the crosslinked, high molecular weight network that forms on the silica surface by the combination of small fragments, radicals, and atoms generated in the plasma [14,17]. Nevertheless, it is possible to estimate the number of layers from the acetylene-PD: The number of carbons can be determined from the degree of deposition for a certain area, and assuming that carbons form a continuous graphite-like structure (i.e., a sp2 conjugated structure) on the entire silica surface, the number of layers can be estimated by considering the silica’s CTAB surface area. Based on this approach, the approximate number of carbon layers on the silica surface was calculated to be from 0.6 to 1.9 layers, as shown in Table 1.

#### 2.1.2. Diffuse Reflectance Infrared Fourier Transform Spectroscopy (DRIFTs) Analysis

Fourier Transform InfraRed (FT-IR) analysis was performed on the pure silica and the modified samples. This technique allows for the identification of organic groups by measuring the absorption of infrared radiation within a particular spectrum of wavelengths. The infrared absorption bands are characteristic of specific molecular components and structures. However, it turned out that direct measurement of plasma-treated silica is challenging due to the prominent silica peak, which is hampering the identification of other peaks in that range [9]. Therefore, plasma polymerization was performed on a glass substrate to investigate the plasma layers (samples P3 and SP3).

In the full spectra shown in Figure 2a, hydrocarbon peaks (3100~2800 cm^−1^ and 1450 cm^−1^) are identified. Furthermore, the peak at 1710 cm^−1^ corresponds to the carbonyl functional groups, which can be formed in the presence of H_2_O or O_2_, with acetylene fragmented in the plasma glow [9]. In Figure 2b, the selected region, 3100~2800 cm^−1^, is zoomed in, and the signal is deconvoluted to identify the hydrocarbons’ detailed structure. The spectra of the acetylene plasma layer are the blue trace, the red line is the fitting line for the deconvolution, and individual peaks after deconvolution are differently colored. In this region, =CH_2_ moieties from unsaturated carbon are identified. [9,18]. These measurements show that the unsaturated structure has remained in the plasma polymer layer, which is expected to improve the filler-polymer interaction via a coupling reaction via free sulfur during vulcanization.

The SP3 spectra (Figure 3a) show that hydrocarbon peaks (3100~2800 cm^−1^ and 1450 cm^−1^) are observed throughout the spectra. Moreover, the peak at 1710 cm^−1^ corresponds to carbonylic groups that can be formed in the presence of moisture in the reactor [19]. A signal at 2550 cm^−1^, which corresponds to a thiol (-SH) group, is also identified.

In Figure 3b, the 3100 and 2800 cm^−1^ regions are zoomed in and deconvoluted to identify the hydrocarbon structures. The black trace in the spectrum corresponds to the DAS-plasma-treated layer, while the color traces represent the individual peaks after deconvolution. Several hydrocarbon structures, including unsaturated C=C bonds, are detected in this region [9,18].

In Figure 3c, the region between 2650~2450 cm^−1^ is magnified, and a thiol peak (-SH) is observed at 2550 cm^−1^ [20]. This peak is most likely formed during the plasma polymer deposition of DAS, as the precursor can undergo fragmentation and recombination to form random and irregular structures [17,21]. The -SH moiety can be formed by dissociating DAS and subsequent reactions with hydrogen.

#### 2.1.3. X-ray Photoelectron Spectroscopy (XPS) Analysis

XPS measurements are utilized for the elemental analysis of the pure and plasma-treated silica, as shown in Figure 4. The neat silica showed higher O1s (532 eV) and Si2p (103 eV) peaks. The P1 and P2 silica presented a significantly increased C1s (284 eV) intensity after the plasma-polymerization process, attributed to the polyacetylene coating on the silica surface.

A higher carbon intensity was observed on the SP2 sample than on the SP1 one, resulting from a higher degree of modification by the acetylene-plasma treatment. Furthermore, sulfur atoms of DAS are detected in both samples, SP1 and SP2. SP1 showed a higher S content than SP2, as expected from the higher degree of DAS deposition and as shown in Figure 5, displaying high-resolution spectra of S2p that indicate the presence of two distinct sulfur species. One species is observed at a low Binding Energy (BE) of 162 eV, corresponding to bound sulfur (i.e., R-S-R). The BE for thiols is between 163 and 164 eV [22,23], which correlates well with the results from DRIFTs. No oxidized sulfur species, such as sulfonates (S2p BE > 166 eV), were detected by XPS on any of the samples [23].

#### 2.1.4. Transmission Electron Microscopy (TEM) Analysis

In Figure 6, the elemental analysis via EF-TEM demonstrates that carbon was barely detectable on the untreated silica. The red and green dots on the combined picture represent silicon (Si) and carbon (C) atoms, respectively.

In Figure 7, acetylene-PD-treated silica, P2, showed a dense carbon distribution on the silica surface. It shows that an acetylene-plasma polymerized coating is formed on the silica surface. The boundaries of individual silica particles seemed to have a higher carbon intensity than the core areas. However, this can be caused by a thicker carbon layer crossed by the electron beam at the edges, where the electron beam passes parallel to the layer instead of perpendicular to the planes. It results in a higher number of inelastic interactions, resulting in a higher intensity profile of carbon on the edges of the nanoparticles compared to the core [24].

Figure 8 shows the results for Sample SP1: The carbon (C) and sulfur (S) elemental mapping showed a homogeneous distribution over the silica surface. In addition, the combined image shows that carbon and sulfur, as represented by green and blue colors, respectively, are distributed evenly over the silicon (Si) particles in red.

In contrast, a more dense carbon layer is observed all over the SP2 surface due to the acetylene-PD treatment showing higher deposition, as shown in Figure 9. Sulfur originating from DAS was also detected. Sample SP2 was designed to have dual layers: acetylene-PD covers the whole surface and thus reduces the filler-filler interaction, and DAS-PD covers the acetylene-PD layer to improve the filler-polymer interaction. The additional acetylene deposition compared to SP1 can be observed in these pictures.

### 2.2. Properties of Silica-Filled Rubber Compounds

#### 2.2.1. Mixing Behavior

Plasma-coated silica was incorporated into the NR formulation with and without a silane coupling agent. These compounds were compared with the ones containing uncoated (Series 1) and in-situ TESPD-modified silica (Series 2). The mixing behavior of the silica-filled NR compounds was evaluated by measuring the mixing torque and temperature profile.

Figure 10a shows the variation of the mixing torque as a function of mixing time for the compounds containing differently modified silicas without using a coupling agent. The acetylene plasma-treated silica (P1, P2)-filled compounds exhibited a slightly lower mixing torque when compared to the NU compound containing uncoated silica. Additionally, the mixing torque for the SP1-filled compound was lower than that of the P1 and P2-filled compounds, while the SP2-filled compound, NSP2, did not demonstrate any reduction in torque. However, the reduction in mixing torque observed for the plasma-treated silica-filled compounds was inferior to that observed for those containing silane.

In Figure 10b, the compound NSP2-HS with 2.3 phr silane shows the highest mixing torque, followed by the compounds NU-HS, NP1HS, and NSP1-HS. The compound (NP2-HS), filled with silica P2 with a higher degree of plasma deposition and 2.3 phr of silane, exhibits similar final torque values compared to the in-situ modified reference compound with 4.5 phr of TESPD (NU-FS), the full amount.

It is known that commercially available pre-treated silica has the disadvantage of unmodified silica surfaces being revealed during the mixing process when silica clusters are broken into smaller units [15]. Therefore, an untreated surface of plasma-modified silica is created during the mixing process when the silica clusters break into smaller units. The hypothesis is that the additional silane coupling agent covered the exposed unmodified silica surface during the mixing process, as depicted in Figure 11. Therefore, introducing a silane coupling agent in the acetylene-PD silica-filled NR compounds might improve their processibility and properties. For P2, the addition of 50% TESPD resulted in a compound viscosity similar to the reference compound with 100% TESPD and in-situ modification.

#### 2.2.2. Filler-Filler Interaction

The strain dependence of the storage modulus G′ of the uncured silica-filled NR compounds is compared in Figure 12. ∆G′ is the difference between G′ at 0.56% and 100% of strain, known as the Payne effect, and is shown in Table 2. A high Payne effect implies a strong filler-filler network [25]. The highest Payne effect was observed in compound NU, the untreated silica-filled NR compound. It is well known that mixing silica into NR faces difficulties due to strong filler-filler interactions and poor filler-polymer compatibility [26]. All plasma-modified silica-filled compounds showed a reduction of the Payne effect compared to the compound NU. The P2-filled compound, NP2, has a lower Payne effect than others, indicating that a higher degree of deposition is beneficial for suppressing the filler-filler interaction. However, NSP2, the SP2-filled compound, raises questions about the cause of the relatively high Payne effect. Despite a higher degree of total deposition including acetylene and DAS modifications accounting for 8.2% of the total deposition, SP2 did not significantly reduce filler-filler interaction. This result could be explained by the DAS layer, which may counteract the hydrophobic acetylene layer and increase the polarity of the silica surface. Additionally, the thiol groups in the DAS layer may establish weak hydrogen bonds with the silanol groups on the silica surface, contributing to the observed effects [16,17].

As also reported by Luginsland and Röben [27], using a silane coupling agent substantially decreased the Payne effect of the silica-filled NR compounds compared to those without silane: NUFS versus NU. The ∆G′ of compounds with silane were ranked as follows: NSP2HS > NUHS = NSP1HS > NP1HS > NP2HS = NU-FS, indicating that improving the surface modification suppresses filler-filler interaction. Compound NP2HS presents a ∆G′ similar to NU-FS: Adding half the amount of silane is sufficient to cover the newly revealed surface of the plasma-pre-treated silica during mixing and avoid the reformation of filler-filler interaction.

#### 2.2.3. Silica Filler Dispersion in NR Matrix

TEM visualized the dispersion behavior of the silica filler in the NR matrix. Figure 13a,b show complete filler networks (fillers as darker areas, elastomer as lighter areas) with varying density throughout the whole NU matrix, the untreated silica-filled compound without a coupling agent. It agrees well with the earlier results of the NU compound showing a high Payne effect, thus a high degree of filler-filler interaction.

Using a coupling agent such as TESPD provides good silica dispersion, a low Payne effect, and improved mechanical properties [25,26,28]. In our study, the compound NU-FS, in which the full amount of TESPD (4.5 phr) is used, shows a well-dispersed filler, as seen in Figure 13c,d. Moreover, compound NP2-HS with 2.3 phr of additional TESPD added to the NR/plasma-treated silica-filled compound shows enhanced filler dispersion comparable to the reference compound (NU-FS) having 4.5 phr of TESPD, as shown in Figure 13e,f.

#### 2.2.4. Cure Behavior of PD-Treated Silica-Filled Compounds

The increase in torque, ∆S′ (S′max−S′min), is affected by polymer-polymer, filler-polymer, and filler-filler interactions. The vulcanization curves of the different silica/NR compounds are shown in Figure 14. In the comparison of the system without a coupling agent (Figure 14a), the curve of the pure silica-filled compound (NU) shows a noticeable torque rise at the beginning of the vulcanization process. This behavior is associated with the flocculation of silica due to the polarity difference between silica and polymer [29] and the interaction of the silanol groups on the silica surface. This effect is significantly reduced in the compounds filled with plasma-treated silica. It was expected that plasma-treated silica would exhibit a higher ∆S′ than the compound with untreated silica due to the improved filler-polymer interaction achieved by the formation of sulfur crosslinks between the double bonds in the deposited layer and the polymer. However, Series 1 compounds showed negligible differences in torque increase except for those samples filled with DAS plasma-treated silica (NSP1). It exhibited an even lower torque increase (∆S′), which will lead to deteriorated vulcanizate properties, including mechanical and dynamic mechanical properties.

Previous studies have indicated that the thiol immobilized on the silica surface can undergo a reaction with the accelerator, e.g., N-cyclohexylbenzothiazole-2-sulphenamide (CBS), resulting in the formation of Intermediate-1. This Intermediate-1 can be converted into the highly active Intermediate-2 in the presence of free sulfur and an amine such as diphenyl guanidine (DPG), which can rapidly react with the polymer, ultimately leading to a shorter scorch time [2,28,30]. The lower torque increase observed in the NSP1 compound can be explained by the presence of the thiol in the PD-DAS-deposited layer. The thiol is likely to localize the curative near the filler surface, as illustrated in Figure 15. Consequently, the rubber matrix has a comparatively lower vulcanization efficiency.

NSP2, which also underwent DAS-PD treatment and contains thiol moieties, showed a higher torque increase compared to NSP1. This could be attributed to the lower concentration of thiols in NSP2, which was confirmed through XPS elemental analysis (0.2% compared to 0.7% in NSP1), as presented in Table 3. The lower concentration of thiol would result in less attraction of curatives toward the filler surface, thus having less influence on the curing behavior of the rubber matrix.

The flocculation behavior of silica is reduced in the samples containing TESPD, as shown in Figure 14b. The compound NU-HS, with 2.3 phr of TESPD added, still presents flocculation at the beginning of the measurement. It is explained by an inappropriate ratio between silane and silanol groups on the silica surface for efficient treatment [32]. However, plasma-treatment of silica with 2.3 phr of additional silane (NP1-HS, NP2-HS, NPS1-HS, and NPS2-HS) and NU-FS, the compound with untreated silica and 4.5 phr of silane added in situ, suppresses re-clustering of the filler. This is in accordance with the torque values at the end of the mixing process, showing the similar compatibility of these two different silica filler systems with the rubber matrix. It appears that the interaction between the plasma-treated silica and rubber is less effective than the covalent filler-polymer bonding generated through the coupling reaction between in-situ TESPD-modified silica and the polymer. This is evident as NP2-HS exhibits a lower ∆S′ than NU-FS, the latter presenting the highest torque increase (∆S′), which is expected since the TESPD reacts with the polymer under curing conditions to form rubber-filler bonds [33]. The compounds containing NP1-HS, NP2-HS, and NSP1-HS show a slightly higher torque increase than the compound NU-HS with only 2.3 phr of TESPD: the double bonds in the deposited plasma layer contribute to the rubber-filler coupling via free sulfur. Conversely, NSP1-HS showed the lowest torque increase among the silane systems studied.

#### 2.2.5. Mechanical Properties of PD-Treated Silica-Filled Compounds

The stress-strain curves of the cured silica-filled NR compounds are compared in Figure 16. In Figure 16a, the plasma-treated silica NP1, NP2, and SP2 compounds showed improved mechanical properties compared to the non-treated silica-filled compound. However, no significant improvement was observed from a higher degree of modification of sample P2 compared to P1 in terms of mechanical properties, in accordance with the ∆S′ results and in contrast with the Payne effect results: the crosslinking network here is the main factor determining the mechanical properties. The DAS-PD-treated silica-filled compound, NSP1, showed lower mechanical properties than the other samples due to the lower torque increase, indicating a lower degree of crosslinking. Moreover, adding a coupling agent significantly improves the mechanical properties of the cured compounds compared to the no-silane system, as can be seen when comparing Figure 16a,b.

M100, M300, and tensile strength values are summarized in Table 4. The compound NU-FS showed the best mechanical properties. The combination of plasma-treated silica and additional TESPD, NP1-HS and NP2-HS, improved the mechanical properties. These results suggest that plasma modification increases the compatibility of the filler with the polymer matrix, improving its rheological and mechanical properties. However, the revelation of an unmodified silica surface while mixing the compound with plasma-modified silica, a disadvantage of pre-treated silica in general, is also occurring for plasma-treated silica. In order to overcome this, half the amount of silane can be added, enhancing the filler-polymer interaction by reacting with the new surface of the plasma-modified silica during mixing.

NSP1-HS had better mechanical properties than its counterpart without additional silane. However, its properties were not as good as those of the in-situ TESPD-modified compounds, primarily due to lower crosslink densities, as indicated by the torque increase during curing. NSP2-HS did not show any significant improvement compared to NU-HS, while notable improvements in mechanical properties were observed in the acetylene PD-treated silica with additional silane. It can be concluded that the combination of acetylene plasma-treated silica with an additional silane coupling agent can further improve the material’s properties.

#### 2.2.6. Dynamic Mechanical Properties PD-Treated Silica-Filled Compounds

Tan δ values are obtained in a temperature sweep. Usually, a higher tan δ at 0 °C indicates better wet grip performance, referring to the braking ability of tires. Lower tan δ values at 60 °C, another lab-scale indicator of tire performance, represent improved rolling resistance (RR), referring to reduced fuel consumption associated with energy-saving tires.

In Table 5, the tan δ at 0 °C values are compared. For easier comparison, all values are normalized relative to the reference value of NU, and indices are used: a higher index signifies a better property. The sample containing NU shows a lower tan δ at 0 °C value due to strong filler-filler networking, resulting in more occluded or trapped rubber and thus reducing the effective rubber fraction, causing hysteresis. On the other hand, plasma-treated silica-filled compounds except for NSP1, i.e., NP1, NP2, and NSP2, showed higher tan δ values at 0 °C value: approximately 9~15% better than the NU sample due to the lower filler-filler interaction, which leads to the destruction and reconstruction of the filler network, causing energy dissipation. This positive effect is remarkable on the surface’s coverage with acetylene-PD treatment in the range investigated here.

Using additional silane reduces filler-filler interaction and increases filler-polymer interaction, which can improve the dynamic properties of the compound. The tan δ values at 0 °C of the compounds NU-FS (having 4.5 phr of TESPD), NU-HS (2.3 phr of TESPD), and NP2-HS (2.3 phr of TESPD) are all higher than the ones of the compounds without silane, which suppresses the filler network formation. NP1-HS, NSP1-HS, and NSP2-HS show a relatively lower value.

The values and indices of tan δ at 60 °C, a lab-scale indicator of a tire’s rolling resistance (RR) performance, are compared and summarized in Table 6. This value should be low for energy-saving tires to decrease the hysteresis loss of tires associated with fuel consumption. The plasma-treated silica-filled compounds without silane coupling agents show higher tan δ values at 60 °C than the unmodified silica-filled compound (NU). The interaction of plasma-treated surfaces, van der Waals interaction, is relatively weak compared to the hydrogen bonding of a pure silica surface. Therefore, the filler network breaks easily and reforms during the measurement, leading to a higher hysteresis than the compound filled with only unmodified silica [34,35].

The addition of a silane coupling agent results in lower tan δ values at 60 °C, indicating improved RR of a tire compound. The tan δ at 60 °C of compounds with silane is ranked as follows: NU-FS < NU-HS < NP2-HS < NP1-HS, indicating crucial improvement by combining additional silane and Acetylene-PD treatment. DAS-plasma-treated silica-filled compounds in both series showed higher tan delta values than the reference compounds: it does not perform well in NR due to the formation of thiol moieties, which causes an inefficient crosslinking of the rubber matrix [6]. This, in turn, results in a high number of free polymer chains, contributing to energy dissipation.

## 3. Discussion

Plasma-polymer-coated silica samples were prepared in a vertical tubular reactor with various precursors in an attempt to improve their compatibility with NR compounds. The degree of modification of the plasma-coated sample, from 2.2% to 8.2%, and the theoretical number of deposited layers, from 0.6 to 1.9, depend on the conditions and are determined based on TGA measurements. In addition, DRIFTs and XPS analysis indicated that hydrocarbon species, including unsaturated carbon structures, are present on the silica surface. Anyway, the modified filler surface contained thiol groups generated from the DAS plasma modification.

The in-rubber tests of NR/silica-filled compounds without additional silane coupling agents indicated that acetylene plasma-treated silica leads to improved compound properties when compared to untreated silica in silane-free compounds. Specifically, plasma treatment resulted in lower filler-filler interaction, corresponding to improved filler dispersion. Furthermore, the mechanical properties and wet traction as determined by the indicator tan δ at 0 °C were enhanced compared to untreated silica-filled NR compounds. However, tan δ at 60 °C, an indicator for RR, deteriorated, as evidenced by higher hysteresis under deformation of the filler network. DAS-PD-treated silica led to lower mechanical and dynamic mechanical properties, particularly for the indicator for rolling resistance. This rendered the DAS-PD treatment unsuitable for reinforcing the NR/silica compound due to thiol functional groups forming within the PD-polymer.

## 4. Materials and Methods

### 4.1. Materials for PD

Highly dispersible silica with a specific surface area measured by Cetyl Trimethyl-Ammonium Bromide (CTAB) adsorption of 177 m^2^/g (ULTRASIL^®^ 7005, Evonik AG, Essen, Germany) was employed as a substrate for the surface treatment. Acetylene as a precursor was purchased from Linde Gas Benelux BV (≥99.6%, Schiedam, The Netherlands). The S-containing precursor, Diallyl sulfide (DAS), was purchased from Sigma-Aldrich (>97%, Zwijndrecht, The Netherlands) as shown in Table 7.

### 4.2. Preparation of PD-Treated Silica

The reactor consists of a Pyrex cylinder 320 mm in length and 150 mm in diameter with a round bottom. The powdery materials to be coated are placed at the bottom of the glass vessel and stirred with a magnetic stirrer for homogeneous treatment. The plasma is generated through an external copper coil with eight turns around the chamber. It is ignited by Radio Frequency (RF) energy generated by an ELITE 600FD-01 (600 W, MKS Instruments, München, Germany) with a matching unit (MKS Instruments, type MWH-5.01, power range of 40–500 W at 13.56 MHz). A Mass Flow Controller (MFC, GE50A, 50 sccm max, MKS Instruments, München, Germany) controls the flow rate of acetylene gas. The DAS input line with a needle valve was added to the reactor configuration. In addition, a dip tube is installed for the monomer to improve contact with the silica. The dimensions are 10 mm in diameter and 250 mm in length, and the end of the inlet reaches the surface of the silica powder. The vacuum outlet is connected to the top lid. This configuration is expected to provide sufficient residence time for the acetylene gas activated by the plasma to react with the silica surface. The design of the reactor setup is illustrated in Figure 17.

The silica was dried in a convection oven at 120 °C for 2 h before use. Next, 40 g of dried silica were placed in the glass reactor. The reactor was vacuumed to a pressure below 0.4 mbar, and various plasma conditions were applied for samples P1 and P2 by varying RF power, acetylene flow rate, and treatment time. In the case of DAS, 40 g of dried silica was charged into the reactor, it was pumped down to 0.4 mbar, and DAS vapor was introduced. The operating pressure was adjusted by varying the DAS vapor flow through a needle valve for DAS-PD treatment. SP1 is treated by DAS only; the plasma treatment of sample SP2 was such that a dual-layered deposition was formed. The sulfur-containing layer was deposited after the acetylene treatment and thus at the outermost surface of the silica clusters. The conditions for preparing the plasma-treated silica samples are presented in Table 8.

### 4.3. Characterization of PD-Treated Silica

Thermogravimetric analysis (TGA 5500, TA Instruments, New Castle, DE, USA) was used to determine the degree of deposition. It was measured from 30 °C to 850 °C with a 20 °C/min ramping rate in an air atmosphere. The TGA weight loss curves were normalized at 200 °C, assuming all physically absorbed water was removed. Therefore, only the weight loss from 200 °C to 850 °C was considered.

FT-IR spectra were taken with a Perkin-Elmer Spectrum 100 series equipped with a Diffuse Reflectance Infrared Fourier Transform spectroscopy (DRIFTs) accessory in the 4000–600 cm^−1^ region. Potassium bromide (KBr, FT-IR grade, ≥99%, Sigma-Aldrich, Zwijndrecht, The Netherlands) was used as baseline and sample preparation reference material. KBr was ground and mixed with the silica samples, which were added at a concentration of 10 wt. % to avoid peak intensity saturation during the measurement. Spectra were collected at a nominal resolution of 4 cm^−1^ with 128 sample scans.

Photoelectron spectra were obtained using a Quantera SXM (Scanning XPS Microprobe, Chanhassen, MN, USA) from Physical Electronics. The spectrophotometer was operated under a vacuum of 3·10^−8^ Torr. A monochromatic Al Kα source (50 W, 1.49 keV) was used for primary excitation at an angle of 45° relative to the sample surface. Survey scans were made to see the gross overall atomic content of the surface layer; they were recorded in the range of 0 to 1200 eV. The measured intensities were in the range of 600 to 12,000 counts s^−1^. The atomic concentrations were calculated by Equation (1).
(1)$$C_{x}=\frac{I_{x}}{S_{x}}/\sum_{i}^{n}\frac{I_{i}}{S_{i}}$$
where *I_i_* is the area of a photoelectron peak and *S_i_* is the relative sensitivity factor of the peak. *C_x_* is the fraction of element *x*, *I_x_* is the peak area of element *x*, *S_x_* is the relative sensitivity parameter, Σ is the sum of all elements, and *n* is the number of elements.

TEM (Transmission Electron Spectroscopy) and EF-TEM (Energy-Filtered TEM) were performed in a Philips 300ST-FEG TEM at an acceleration voltage of 300 kV. For the measurements, 0.1 g of silica was dispersed in ethanol by ultrasonication to prepare a suspension. Then, a droplet (0.2 μL) of this suspension was put on a Holey carbon film on a TEM grid, and ethanol was evaporated. A GATAN Ultrascan 1000 (2k × 2k CCD) camera was used for imaging.

### 4.4. Preparation and Evaluation of PD-Treated Silica-Filled NR Compound

The plasma-treated silica samples were tested in a NR/silica formulation in order to evaluate their in-rubber properties. The series was split into two parts:

(1)Series 1 (No Silane):

A reference sample of 55 phr of untreated silica was employed. The amount of plasma-treated silica was adjusted according to the degree of deposition measured by TGA for equal inorganic content in all samples.

(2)Series 2 (+Silane):

The influence of an additional silane coupling agent was investigated. For the references, 4.5 phr of TESPD and 2.3 phr of 50% reduced TESPD are applied to the compounds with 55 phr of untreated silica. In addition, 2.3 phr TESPD was added to the compound filled with plasma-modified silica. The detailed formulations are shown in Table 9 and Table 10.

The NR compounds were prepared in a Plastograph internal mixer (Brabender, Duisburg, Germany). The capacity of the mixing chamber was 50 mL, and the fill factor was 70%. A two-step mixing process was applied, as described in Table 11.

The Payne effect of the silica-filled rubber compounds is well known to indicate the degree of filler-filler interaction. The uncured compoundS′ storage shear moduli (G′) were evaluated using a Rubber Process Analyzer (RPA; RPA Elite, TA Instruments, New Castle, DE, USA). The Payne effect of the uncured rubber was measured at a temperature of 100 °C, a frequency of 1.67 Hz, and varying strains in the range from 0.56 to 200%. The values were calculated from the difference in storage shear moduli at strains of 0.56% and 100%, i.e., G′ (0.56%)–G′ (100%).

In order to determine the vulcanization time, rheograms were taken using the same Rubber Process Analyzer at 150 °C for 30 min under 0.5 degrees (~7%) of strain and at a frequency of 1.667 Hz according to ASTM D5289-95. As vulcanization time, t_90_ + 2 min was used for all samples. Vulcanization of the rubber compound was performed in a Wickert laboratory press (WLP1600, Landau, Germany).

TEM was used to investigate the filler dispersion in the NR/silica vulcanizates. First, a piece of approximately 10 × 5 mm^2^ was cut from the middle part of a rubber sample using a razor blade. Next, the samples were trimmed to a surface of approximately 200–300 μm using a diamond tool, and subsequently, ultrathin sections (70 nm) were obtained at −60 °C using a Leica EM UC7/FC7 ultramicrotome equipped with a Diatome© 35° knife. Finally, the sections were put on a Cu200C copper grid with a carbon support layer. The TEM samples were studied on an FEI Tecnai 20 (Sphera, FEI Company, Hillsboro, OR, USA) operated at 200 kV, equipped with a LaB6 filament, and a Ceta 4k camera to obtain several images at various magnifications.

According to ISO 37, type 2 dumbbell test specimens of the cured rubber compounds were prepared. The tensile properties (i.e., moduli at different strains, tensile strength, and elongation at break) were tested with a Zwick tensile tester model Z1.0/TH1S (Zwick Roell Group, Ulm, Germany) at a cross-head speed of 500 mm/min.

The viscoelastic properties were determined using a Dynamic Mechanical Analyzer (DMA, GABO EPLEXOR^®^, Netzsch, Selb, Germany) in a temperature sweep in tension mode at 10 Hz. First, 0.1% dynamic and 0.3% static strain were applied in the temperature range from −80 °C to 25 °C with a 2 °C/min heating rate to obtain an indicator of the wet grip of tire treads. Then, the strain was switched to 3% dynamic and 5% static within the temperature range up to 80 °C with a 2 °C/min heating rate to obtain the indicator for rolling resistance.

## 5. Conclusions

A reduced quantity of the silane coupling agent added to the acetylene plasma-treated silica-filled compounds improved the compound’s properties. In addition, the silane modifies the fresh surface of silica revealed due to cluster breakdown during mixing. Most properties of the compound with acetylene plasma-treated silica and 50% reduced silane were comparable to the in-situ silane-modified reference compound containing a 100% coupling agent, with the exception of the tan delta at 60 °C, which was slightly inferior to the reference compound. This facilitates processing as lower amounts of volatile organic compounds, such as ethanol, are generated compared to the conventional silica-silane filler systems.

## Figures and Tables

**Figure 1 molecules-28-06646-f001:**
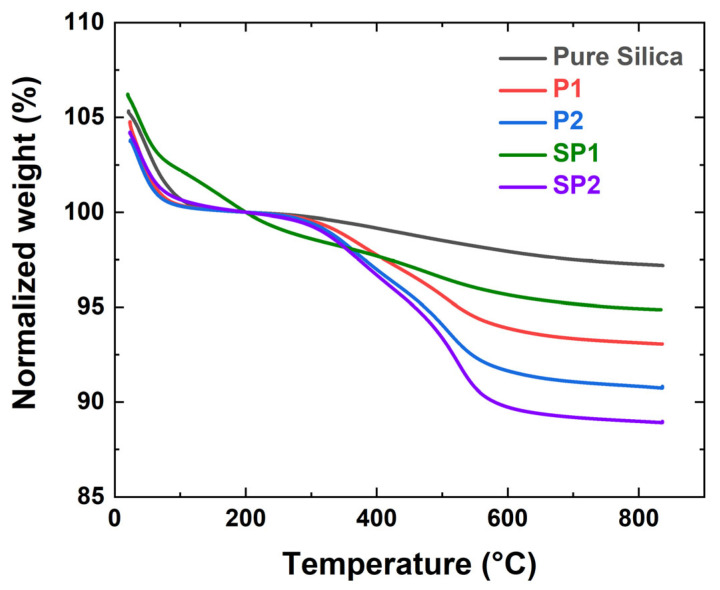
Normalized TGA curves of pure and plasma-treated silica.

**Figure 2 molecules-28-06646-f002:**
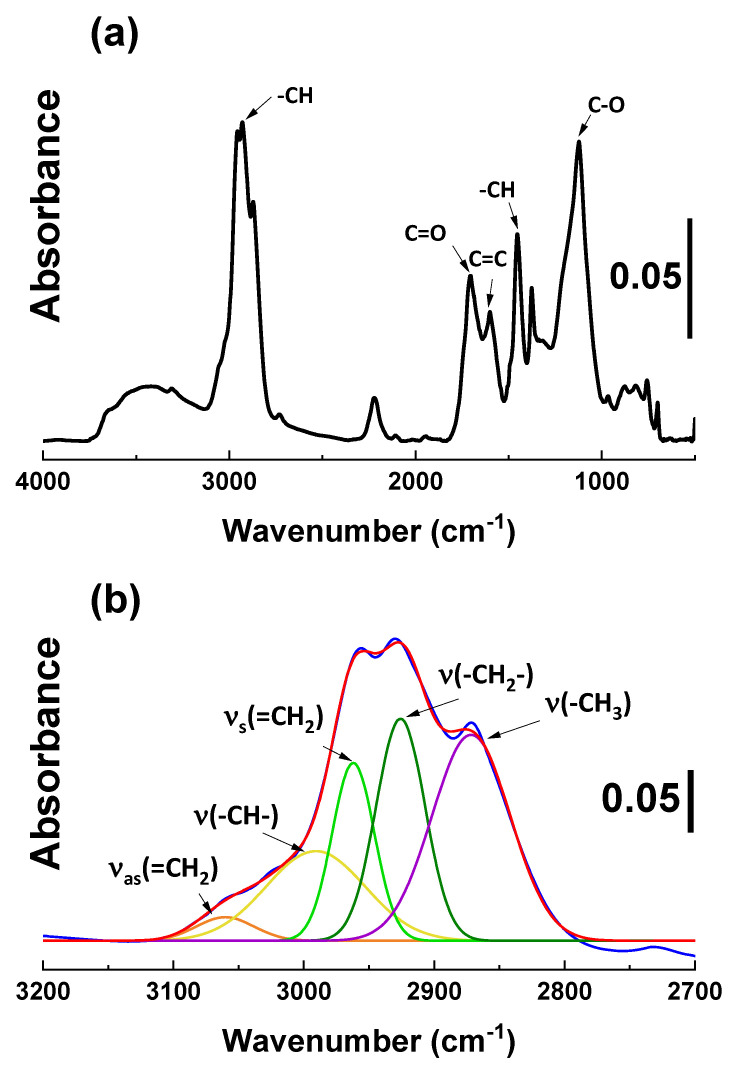
DRIFTs spectra of P3: (**a**) full spectra; (**b**) selected region 3150~2800 cm^−1^: plasma-treated layer (blue trace) and individual peaks after deconvolution (color traces).

**Figure 3 molecules-28-06646-f003:**
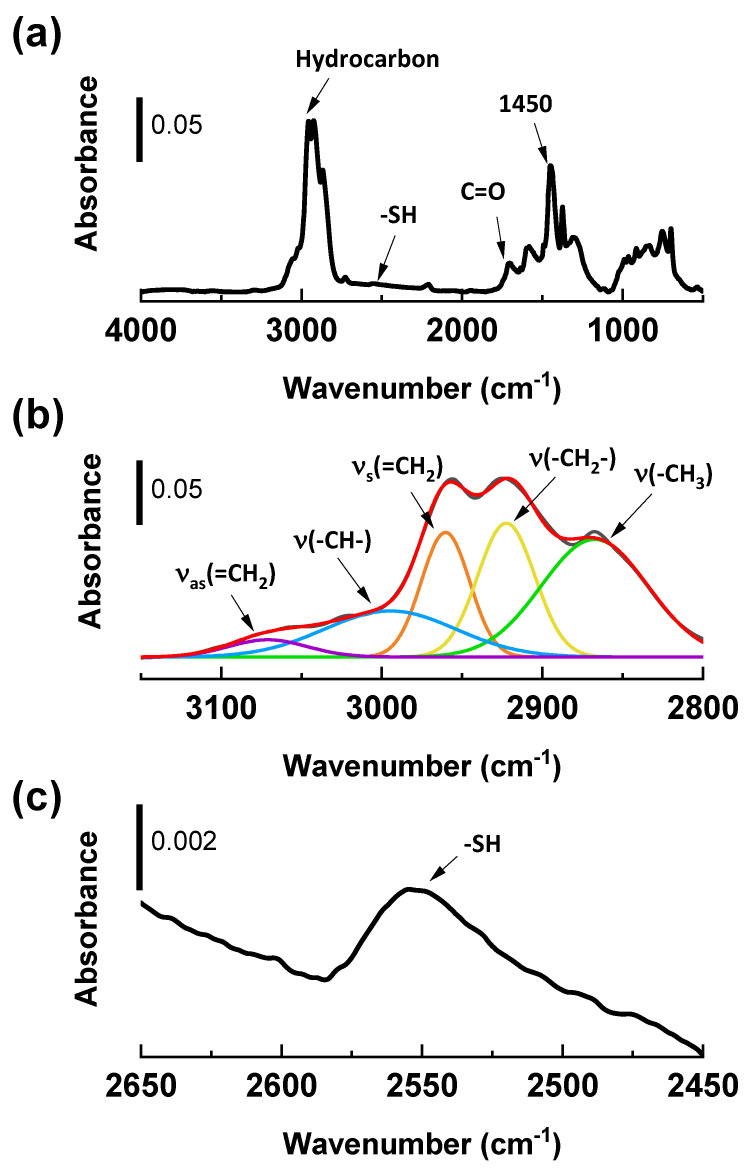
DRIFTs spectra of SP3, DAS-plasma treated layers: (**a**) full spectra; (**b**) selected region of DAS-plasma deposited layer, 3150~2800 cm^−1^ (black trace) and individual peaks after deconvolution (colored traces); (**c**) selected region, 2450~2650 cm^−1^, to identify the -SH moiety [18,20].

**Figure 4 molecules-28-06646-f004:**
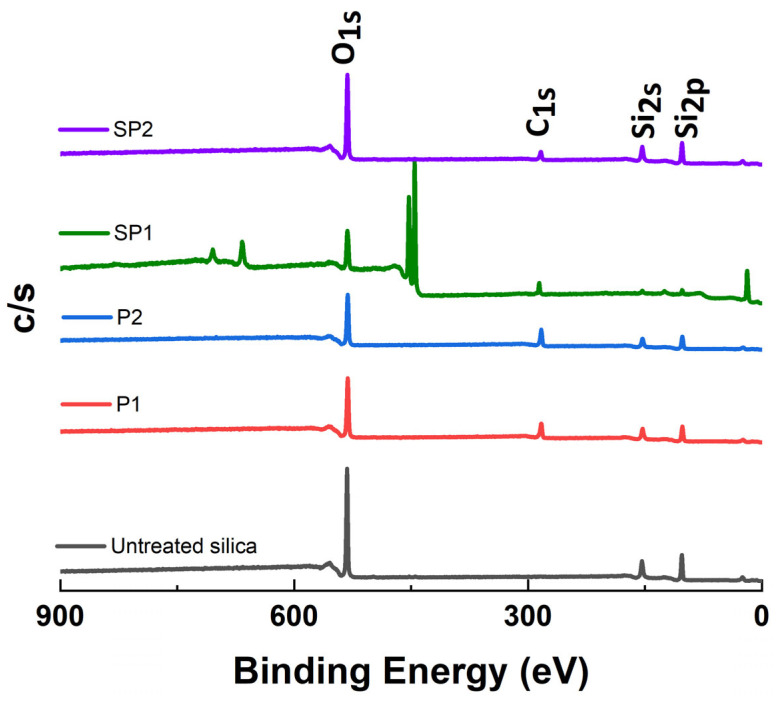
XPS results of pure and plasma-treated silica.

**Figure 5 molecules-28-06646-f005:**
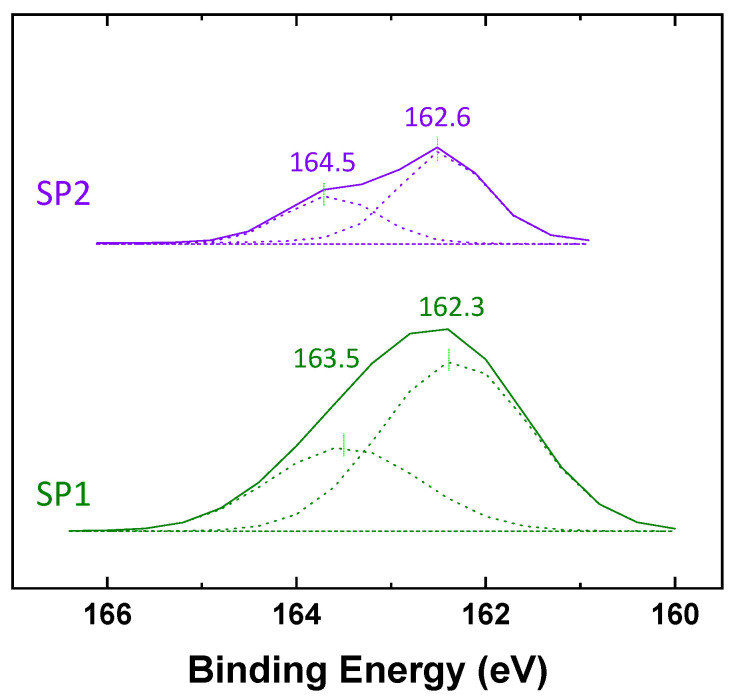
XPS spectra in the binding energy range of 160~166 eV for determination of the sulfur type in the plasma-polymerized DAS (SP1, SP2).

**Figure 6 molecules-28-06646-f006:**
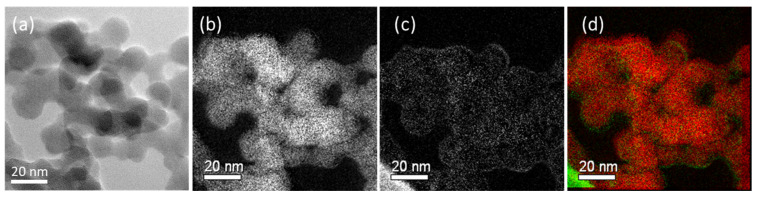
TEM and Energy Filtering TEM images of untreated silica: (**a**) TEM image; (**b**) silicon mapping; (**c**) carbon mapping; (**d**) combined silicon and carbon (Si: red, C: green).

**Figure 7 molecules-28-06646-f007:**
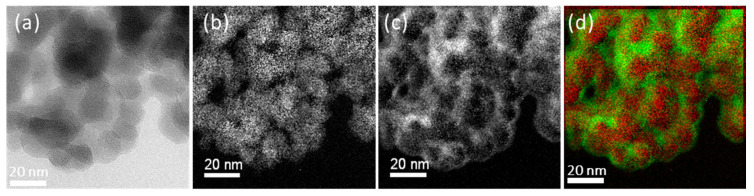
TEM and Energy Filtering TEM images of P2: (**a**) TEM image, (**b**) silicon mapping, (**c**) carbon mapping, and (**d**) combined silicon and carbon (Si: red, C: green).

**Figure 8 molecules-28-06646-f008:**
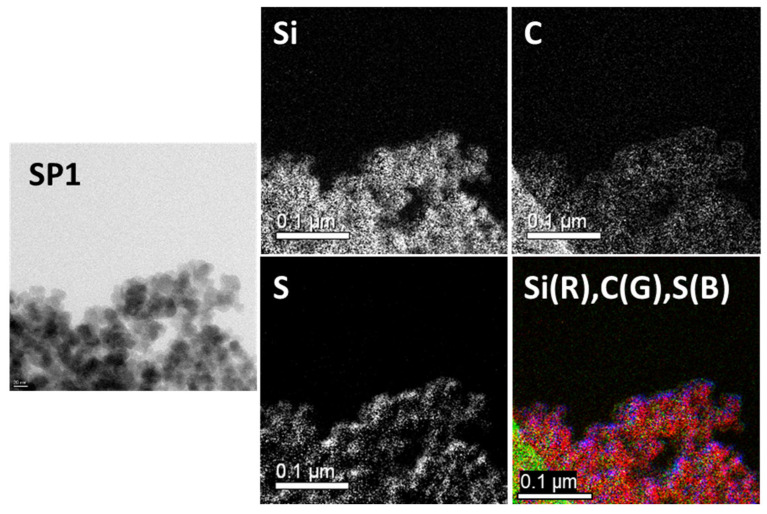
TEM (**left**) and EF-TEM (**middle** and **right**) images of Sample SP1 (Si: red, C: green, S: blue).

**Figure 9 molecules-28-06646-f009:**
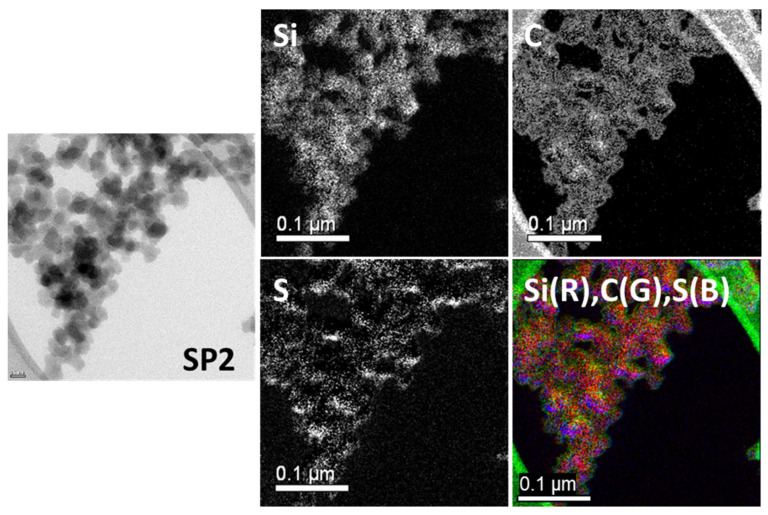
TEM (**left**) and EF-TEM (**middle** and **right**) images of Sample SP2 (Si: red, C: green, S: blue).

**Figure 10 molecules-28-06646-f010:**
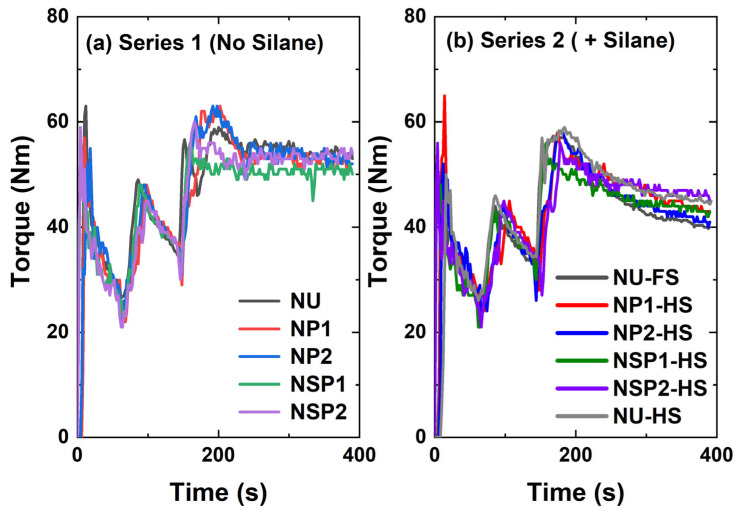
Mixing torque profiles of silica-filled NR compounds.

**Figure 11 molecules-28-06646-f011:**
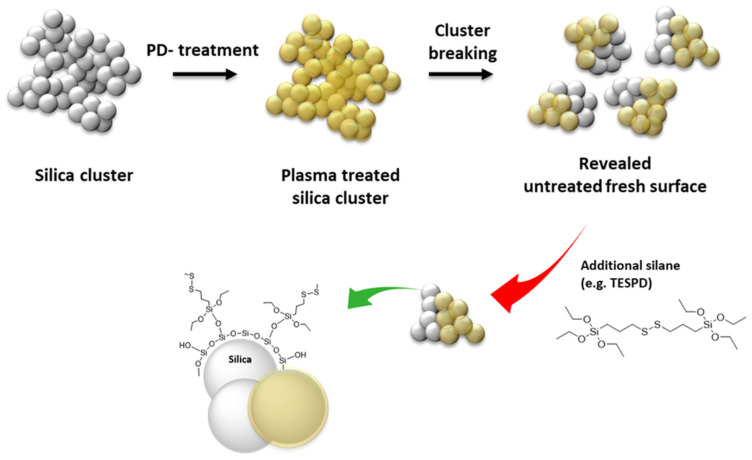
Hypothesis: Reaction of additional silane for the use of acetylene plasma-coated silica in the rubber compound.

**Figure 12 molecules-28-06646-f012:**
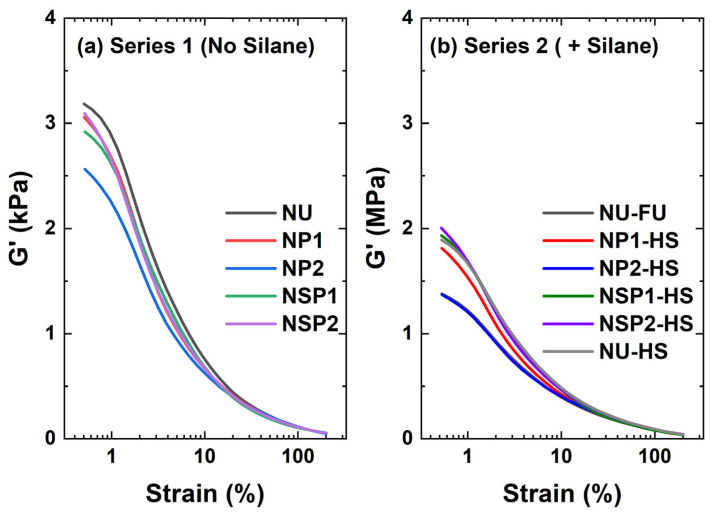
Strain dependency of shear modulus (filler-filler interaction, uncured).

**Figure 13 molecules-28-06646-f013:**
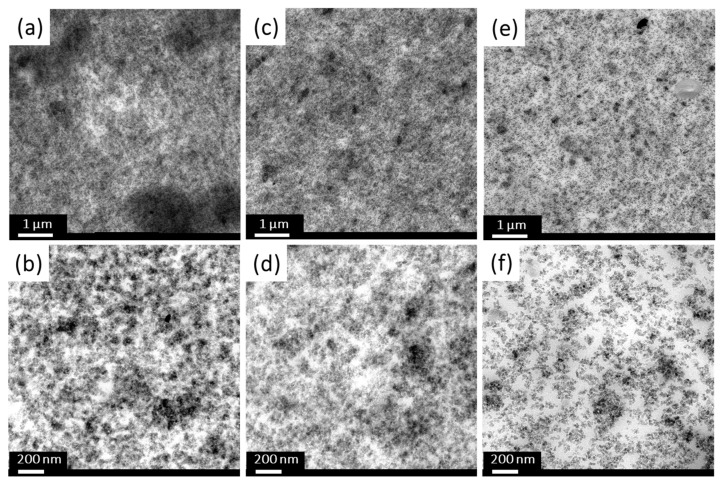
TEM images of NR/silica-filled compounds: (**a**,**b**) NU, (**c**,**d**) NU-FS, (**e**,**f**) NP2-HS.

**Figure 14 molecules-28-06646-f014:**
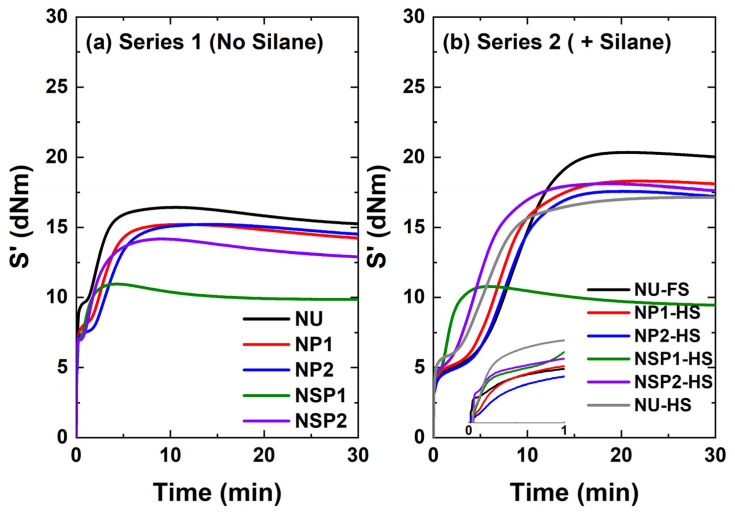
Vulcanization curves of silica-filled NR compounds.

**Figure 15 molecules-28-06646-f015:**
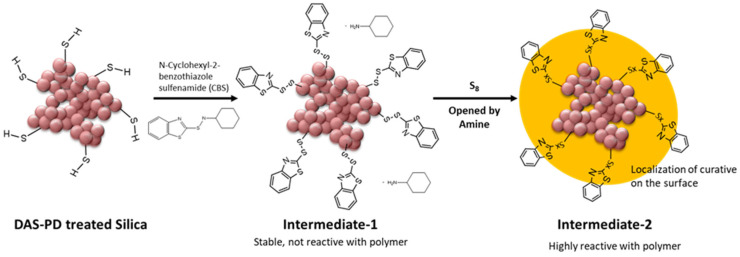
Hypothesis: Localization of curatives on the DAS-PD treated silica surface [31].

**Figure 16 molecules-28-06646-f016:**
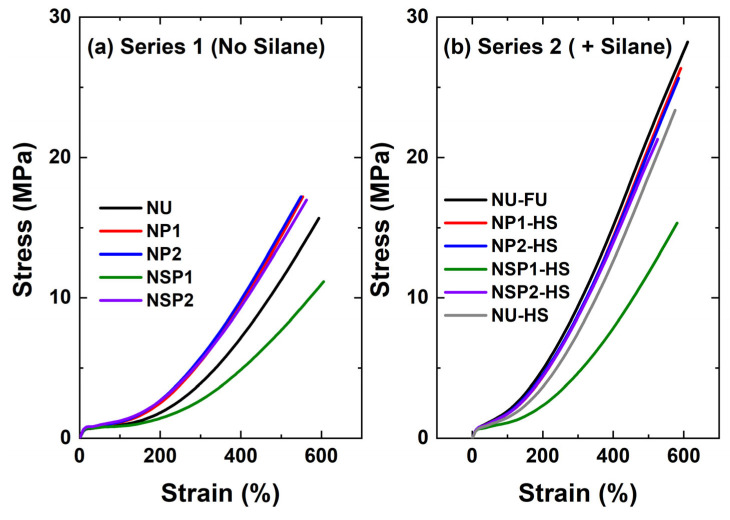
Stress-strain curves of silica-filled NR compounds.

**Figure 17 molecules-28-06646-f017:**
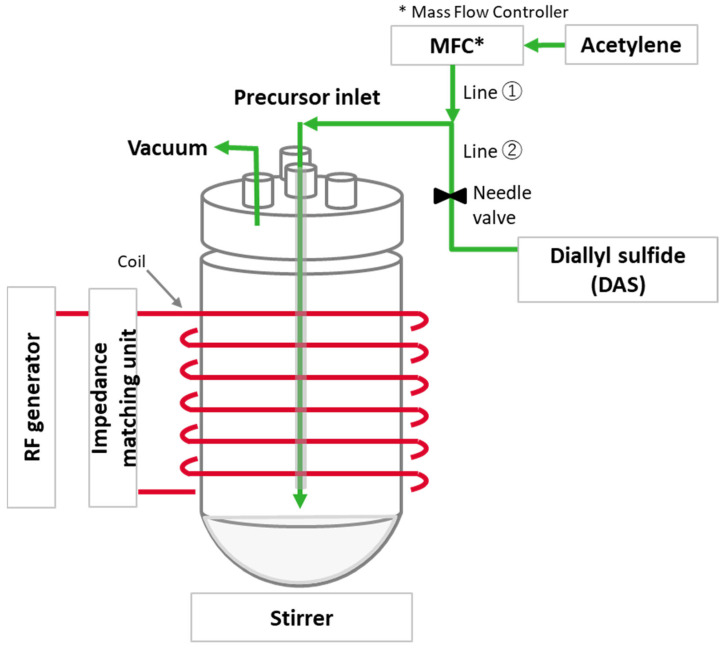
Tailor-made plasma reactor for modification.

**Table 1 molecules-28-06646-t001:** Degree of deposition from the TGA measurement.

Sample Code	Weight Change (200~850 °C) (%)	Degree of Deposition (%)	Estimated Number of Deposited Carbon Layers
Pure silica	2.9 ± 0.1	-	-
P1	7.0 ± 0.1	4.1 ± 0.1	1.1
P2	9.3 ± 0.2	6.5 ± 0.2	1.5
SP1	5.1 ± 0.1	2.2 ± 0.1	0.6
SP2	11.1 ± 0.1	8.2 ± 0.1	1.9

**Table 2 molecules-28-06646-t002:** ∆G′ of the PD-treated silica-filled NR compounds.

Sample	Series 1 (No Silane)					Series 2 (+Silane)					
	NU	NP1	NP2	NSP1	NSP2	NUFS	NP1HS	NP2HS	NSP1HS	NSP2HS	NUHS
∆G′_0.56–100%_ (MPa)	3.1	2.9	2.4	2.8	3.0	1.3	1.7	1.3	1.8	1.9	1.8

**Table 3 molecules-28-06646-t003:** XPS results of untreated silica, SP1, and SP2 plasma-treated silica.

Sample	Elemental Composition			
	C	O	Si	S
	%	%	%	%
Pure silica	1.1 ± 0.5	69.8 ± 0.5	29.1 ± 0.9	-
SP1	6.3 ± 0.7	65.4 ± 0.6	27.6 ± 0.3	0.7 ± 0.1
SP2	15.8 ± 1.2	59.2 ± 0.6	24.8 ± 0.6	0.2 ± 0.1

**Table 4 molecules-28-06646-t004:** Mechanical properties of silica-filled NR compounds.

Compound Code		M100	M300	Tensile Strength	Elongation at Break
		MPa	MPa	MPa	%
Series 1 (No silane)	NU	1.0 ± 0.1	3.9 ± 0.2	15.7 ± 1.0	593 ± 20
	NP1	1.1 ± 0.1	5.5 ± 0.2	17.2 ± 1.5	554 ± 21
	NP2	1.2 ± 0.1	5.7 ± 0.1	17.2 ± 0.9	549 ± 20
	NSP1	0.9 ± 0.1	2.7 ± 0.3	11.2 ± 0.7	605 ± 24
	NSP2	1.2 ± 0.1	5.5 ± 0.2	17.0 ± 1.9	563 ± 33
Series 2 (+silane)	NU-FS	1.9 ± 0.1	9.4 ± 0.4	28.2 ± 0.9	611 ± 24
	NP1-HS	1.7 ± 0.1	8.7 ± 0.4	26.4 ± 0.6	592 ± 11
	NP2-HS	1.8 ± 0.1	8.8 ± 0.5	25.7 ± 1.5	586 ± 26
	NSP1-HS	1.1 ± 0.1	4.7 ± 0.2	15.3 ± 0.6	581 ± 19
	NSP2-HS	1.7 ± 0.1	8.7 ± 0.3	21.3 ± 0.7	526 ± 13
	NU-HS	1.5 ± 0.1	7.5 ± 0.6	23.4 ± 2.6	575 ± 32

**Table 5 molecules-28-06646-t005:** Wet grip indicator (tan δ @ 0 °C) comparison of silica-filled NR compounds.

Sample	Series 1 (No Silane)					Series 2 (+Silane)					
	NU	NP1	NP2	NSP1	NSP2	NUFS	NP1HS	NP2HS	NSP1HS	NSP2HS	NUHS
Tan δ @ 0 °C	0.092	0.105	0.106	0.084	0.100	0.115	0.104	0.113	0.104	0.106	0.114
Index	100	114	115	92	109	125	113	123	113	116	124

**Table 6 molecules-28-06646-t006:** RR indicator (tan δ @ 60 °C) comparison of silica-filled NR compounds.

Sample	Series 1 (No Silane)					Series 2 (+Silane)					
	NU	NP1	NP2	NSP1	NSP2	NUFS	NP1HS	NP2HS	NSP1HS	NSP2HS	NUHS
Tan δ @ 60 °C	0.169	0.193	0.188	0.219	0.194	0.142	0.161	0.153	0.220	0.184	0.150
Index	100	86	89	70	85	116	105	109	70	91	111

**Table 7 molecules-28-06646-t007:** Precursors for plasma modification.

Precursor	Molecular Formula	Molecular Weight (g/mol)	Boiling Point (°C)	Vapor Pressure @ 20 °C
Acetylene	C_2_H_2_	26	−84	-
Diallyl sulfide (DAS)	C_6_H_10_S	114	138	9 mbar

**Table 8 molecules-28-06646-t008:** Experimental conditions employed for the plasma modification of the silica substrate.

Sample ^(1)^	Silica (g)	RF Power (W)	Precursor		Treatment Time (h)
			Type	Monomer Pressure	
P 1	40	300	Acetylene	30 sccm ^(2)^	12
P 2	40	450	Acetylene	50 sccm	12
SP1	40	450	DAS	0.6 mbar ^(3)^	12
SP2	40	450	1st-Acetylene	50 sccm	10
			2nd-DAS	0.6 mbar	2
P 3 ^(4)^	-	450	Acetylene	50 sccm	2
SP3 ^(4)^	-	450	DAS	0.6 mbar	2

^(1)^ S stands for a sulfur-containing precursor (i.e., DAS), and P for plasma treatment. ^(2)^ Standard cubic centimeters per minute (sccm); the flow rate of acetylene gas was controlled via a mass flow controller (MFC). ^(3)^ The needle valve controls the vapor flow of DAS to make 0.6 mbar of reactor pressure, corresponding to the feed rate of 3.1 g ± 0.2/h of DAS vapor. ^(4)^ A plasma polymerized layer on a glass substrate instead of on the silica surface was performed for the DRIFTs (Diffuse Reflectance Infrared Fourier Transform Spectroscopy) measurement due to the difficulties with this method when using silica particles.

**Table 9 molecules-28-06646-t009:** NR formulations for Series 1 with plasma-treated silica.

Mixing	Sample Code ^(1)^	Series 1 (No Silane)				
		NU	NP1	NP2	NSP1	NSP2
Step 1	NR (SIR 20)	100	100	100	100	100
	Silica	55	-	-	-	-
	P1	-	57.3	-	-	-
	P2	-	-	58.5	-	-
	SP1	-	-	-	56.2	-
	SP2	-	-	-	-	59.5
	ZnO/SA/6PPD /TDAE Oil	2.5/1.0/2.0/8.0				
Step 2	Sulfur/CBS/DPG	1.4/1.7/1.0				

Unit: parts per hundred of rubber (phr); ^(1)^ N: Natural Rubber (SIR 20); U: Untreated silica; P1, P2, SP1, and SP2: Plasma-treated silica.

**Table 10 molecules-28-06646-t010:** Series 2 formulations with additional silane for in-rubber tests of plasma-treated silica.

Mixing	Sample Code ^(1)^	Series 2 (+Silane)					
		NUFS	NP1HS	NP2HS	NSP1HS	NSP2HS	NUHS
Step 1	NR (SIR 20)	100	100	100	100	100	100
	Silica	55	-	-	-	-	55
	P1	-	57.3	-	-	-	-
	P2	-	-	58.5	-	-	-
	SP1	-	-	-	56.2	-	-
	SP2	-	-	-	-	59.5	-
	TESPD	4.5	2.3	2.3	2.3	2.3	2.3
	ZnO/SA/6PPD /TDAE Oil ^(2)^	2.5/1.0/2.0/8.0					
Step 2	Sulfur/CBS/DPG	1.4/1.7/1.0					

Unit: parts per hundred of rubber (phr); ^(1)^ N: Natural Rubber (NR); U: Untreated silica; P1, P2, SP1, and, SP2: Plasma treated silica; see Table 8. FS: Full amount of silane (TESPD 4.5phr), HS: Half amount of silane (TESPD 2.3phr); ^(2)^ ZnO: Zinc oxide, SA: Stearic acid, 6PPD: N-(1,3-dimethyl butyl)-N’-phenyl-p-phenylenediamine, TDAE Oil: Treated Distillate Aromatic Extracted Oil.

**Table 11 molecules-28-06646-t011:** Mixing procedures of NR/silica compounds.

Step 1, initial conditions: 80 °C and 70 rpm	
Time (mm: ss)	Action
0:00	The addition of rubber, mastication
1:20	Addition of 1/2 filler and silane (for Series 2)
2:40	Addition of 1/2 filler, TDAE Oil, and chemicals
3:50	Adjusted rotor revolution to keep the temperature at 150 °C
6:30	Discharge
Step 2, initial conditions: 60 °C and 60 rpm (24 h. after step 1)	
Time (mm: ss)	Action
0:00	Addition of masterbatch from Step 1, mastication
1:00	Addition of curatives (sulfur, CBS ^(1)^, and DPG ^(2)^)
2:40	Discharge

^(1)^ CBS: N-Cyclohexyl-2-benzothiazole sulfenamide, ^(2)^ DPG: 1,3-Diphenylguanidine.

## Data Availability

The data presented in this study are available on request.

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
