# Peer review of "Plasma Polymerization of Precipitated Silica for Tire Application"

_molecules, 2023, doi:10.3390/molecules28186646_

Round 1

Reviewer 1 Report

Review of molecules-2609322

This is an interesting submission of plasma polymerization on powder, which is quite challenging. The treated powdery samples were properly characterized physically and chemically, as well as the application of the treated samples for improving tire performance. There are several issues that must be addressed in the revised manuscript, as follows:

  1. Abstract: Please state the potential of this study.
  2. Please add the Conclusion section.
  3. Please add list of abbreviations.
  4. Please write “diallyl” without separation in all part of the manuscript, and Figure 17.
  5. What is “phr” unit?? Please define clearly.
  6. Introduction: Many references are from early 2000s. Please add more references from 2020s.
  7. Line 122: =CH2 --> subscripted 2
  8. Figure 3c vs 3a: The –SH peak is very very tiny! It is overpowered with other peaks in the region of 2800-3100 cm-1. It is not very convincing. Please repeat the experiment to obtain strong data.
  9. Figure 3a: Please write the related moieties/functional groups instead of indicating the wavenumber only. Check the appropriate example in Figure 2a.
  10. Can the treatment duration reduced to 1-2 hours while still giving homogeneously treated samples?

  1. Table 7: C2H2, C6H10S --> Please use proper subscripted numbers for the chemical formula.
  2. Line 456: m2/g --> superscripted 2
  3. Line 472, 493: Please write the unit sccm in all lowercase letters, like those in Table 8.
  4. Line 491: Please do not write the caption of Table 8 in italic
  5. Line 526: 0.2 µL --> use uppercase L for liter, and then add as space between “0.2” and “µL”.
  6. Line 515: 1486 eV --> Delete the space between “148” and “6”. Add a space between “6” and “eV”
  7. Equation 1: Please use appropriate subscripted letters, especially for Si. Without subscripted lowercase i, it will confuse the reader with Si of silicon. Please write Cx, Ix, Sx, Ii, Si with proper subscripted letters.
  8. Line 534: Please do not start a sentence with numbers.
  9. Line 540: …2.3 phr, 50% reduced TESPD…
  10. Line 570: 10×5 mm2 --> Please use multiplication sign, and superscripted number 2.
  11. Line 571: ..of approximately 200-300 µm..
  12. Line 572: -60 °C --> Please delete the space between the dash and “60”.

  1. References 7, 8, 10, 11, 12, 13, 30, 34, 36: Please write the name of the journal correctly --> Rubber Chemistry and Technology --> uppercase R, C, T

Author Response

Dear Reviewers

Thank you very much for reviewing our manuscript entitled ‘Plasma Polymerization of Precipitated Silica for Tire Application’ (article # molecules-2609322). We appreciate your comments and corrections and have adjusted the manuscript accordingly. Please find the detailed responses below and the corresponding corrections highlighted in the re-submitted files.

Best regards,

S.KIM

Reviewer 2 Report

Dear Authors,

your manuscript provides a careful and detailed study of the influence of silica pretreatment by means of plasma deposition of hydrocarbonous layers on the mixing behavior, morphology  and mechanical as well as dynamic-mechanical properties of silica reinforced natrual rubber formulations vice versa "conventional" in-situ silanized or untreated NR-Si reference compounds.

The presentation of methods, results, and their discussion as well as the conclusions drawn are all very well, and I could follow completely your argumentation.

The only a bit "open point" which remained to me is your statement in the lines 217-219 on page 9 "...The SP1-filled compound shows slightly better processability than others,....". This statement refers to the discussion of the mixing torque presented in Fig. 10 only, and I have the question how such a statement on ease/difficulty for processing the compounds -which in my understanding relates than to the shaping/extrusion/calendering behavior of the compound after the mixing- can be concluded form comparing the final mixing torque values (without giving detailed information on the actual compound temperatures!) only? Can you validate this statement with results of other methods (from your RPA-studies, curing curves or other rheometric measurements or even extrusion/Garvey-tests)? Please consider in the discussion.

The above question remains the only open point to me.

Author Response

Dear Review,

Thank you very much for reviewing our manuscript entitled ‘Plasma Polymerization of Precipitated Silica for Tire Application’ (article # molecules-2609322). We appreciate your comments and corrections and have adjusted the manuscript accordingly. Please find the detailed responses below and the corresponding corrections highlighted in the re-submitted files.

Best regards,

S.KIM

Round 2

Reviewer 1 Report

Thank you for the significant improvement applied to this manuscript. It can be accepted now.